# Using an algorithmic approach to shape human decision-making through attraction to patterns

Haran Shani-Narkiss [1] ✉, Baruch Eitam [2] ✉ & Oren Amsalem [3] ✉

Evidence suggests that people are attracted to patterns and regularity. We hypothesized that decision-makers, intending to maximize profit, may be lured by the existence of regularity, even when it does not confer any additional value. An algorithm based on this premise outperformed all other contenders in an international challenge to bias individuals' preferences. To create the bias, the algorithm allocates rewards in an evolving, yet easily trackable, pattern to one option but not the other. This leads decision-makers to prefer the regular option over the other 2:1, even though this preference proves to be relatively disadvantageous. The results support the idea that humans assign value to regularity and more generally, for the utility of qualitative approaches to human decision-making. They also suggest that models of decision making that are based solely on reward learning may be incomplete.

Humans display the ability to learn and adapt to the statistical properties of their environment, whether these are purely perceptual[1] or reward related[2], and often without an explicit goal to do so[3,4]. Such ubiquitous learning of regularity may reflect the attraction to structure or predictability. For example, human attention has been shown to be pulled towards regularity[5–7], and movements with predictable sensory outcomes are reinforced from infancy[8] and throughout life[9].

These seemingly reinforcing properties of regularity may also influence the development of higher-level mental representations such as stereotypes, thereby reinforcing an erroneous interpretation of social structures. Specifically, people were willing to forego payment to encounter stereotype-confirming information (compared to stereotype-violating information), which was shown to activate the brain's reward-related areas[10].

Evidence thus suggests that humans are geared towards consistently searching for—and holding onto—regularities in an ever-changing environment, whether knowingly or not. However, direct evidence that regularity is itself desired or reinforcing, is scarce, not to mention harnessing regularity to actually affect people's choice-behavior or preferences.

Nudging refers to efforts to subtly influence individuals into making subjectively better or socially more desirable decisions. The concept has received much scientific and popular attention[11,12]. A well-known example is the large effect of the default option ("opt out" or "opt in") on the percentage of people willing to donate organs; it was shown that a 16% to 57% increase in such willingness could be achieved simply by using "donate" as the default and "opt out" as the active decision[13]. Since empirical support seemed to be initially strong, as were the apparent cost-benefit ratios, both the UK and US set up specialized teams[14] and government offices to develop these practices. Although both have since been discontinued, other international institutions such as the World Health Organization (WHO) continue to advocate for the use of such interventions[15].

Unfortunately, recent evidence suggests that the impact of these interventions has been overestimated[16–18], and several influential results have been reported to be false[19–22]. A recently published meta-analysis suggests that when publication bias is accounted for, the average effect of nudges is negligible (Cohen's $d = 0.04$ with a 95% credible interval of 0.00–0.14[17]). Understandably, these findings

[1]UCL Sainsbury Wellcome Centre for Neural Circuits and Behaviour, London W1T 4JG, UK. [2]School of Psychological Sciences, University of Haifa, Mount Carmel, Haifa, Israel. [3]Division of Endocrinology, Diabetes and Metabolism, Department of Medicine, Beth Israel Deaconess Medical Center, Harvard Medical School, Boston, MA 02215, USA. ✉e-mail: h.shani-narkiss@ucl.ac.uk; beitam@psy.haifa.ac.il; oren.a4@gmail.com

prompted a reckoning[23], along with doubts as to the soundness of nudging altogether[24].

Nevertheless, a premature shift towards disbelief in the efficacy or feasibility of nudging could prove even costlier than the blanket endorsement of such interventions. This is likely to be especially problematic since the use of nudges may also be deleterious (negative nudges are also referred to as "sludges" or dark patterns)[25,26].

Whether one's ultimate goal is to deploy effective pro-social interventions, to mitigate the harm done by interventions that effectively undermine social interests, or to test whether and how various non-tangible reinforcers influence decision making—the impact of potential interventions might be best measured in well controlled environments.

We estimated the degree to which humans value regularity in and of itself within the framework of the Choice Engineering Competition announced in *Nature Communications* (CEC[27]). Briefly, researchers submitted algorithms for allocating a finite number (50) of constant-size, monetary (1¢) rewards to two possible choice options. The goal of the competition was to covertly influence human participants towards making a particular choice (the bias-towards, or target-side, hereafter referred to as *Bias*+) and away from another (the bias-away, or non-target-side, hereafter referred to as *Bias*−).

The CEC imposed an important constraint, rewards were to be evenly divided between the two options. This made it an excellent testing ground for the influence of potential psychological attractors on repeated decision making, above and beyond that of objective and tangible outcomes.

To test the algorithms' ability to bias participants' choices, the competition organizers recruited 3521 human participants via Amazon Mechanical Turk, each of whom was paired with a single algorithm (see Supplementary Data 1 for the number of participants allocated to each algorithm, and the numerical description of all results and statistics). During the CEC, participants were repeatedly presented with two possible choices, for a total of 100 decisions. All performed the same task; their goal was to collect as many rewards as possible; after each choice they were informed whether they had obtained a reward (see Fig. 1a–c for a depiction of the task). Unbeknownst to them, 25 rewards were evenly allocated to both the Bias+ and Bias− sides (50 in total).

In any given session, a single designated algorithm determined the reward schedule; i.e., when and where would rewards appear throughout the session. This was done in either a deterministic manner, via the competition's "Static track", or dynamically in response to the participant's ongoing choices, via the "Dynamic track". In compliance with the CEC rules, we submitted 2 schedules to the static track (SS2 and SS4) and two algorithms to the dynamic track (DS1 and DS2). Note that the term "schedule" is used to describe the algorithms' output.

The two submissions to the CEC's dynamic track were slightly different variations of an algorithm designed to harness people's apparent tendency to value regularity for biasing their choices. More specifically, the algorithms' design was based on the following premises: (1) exploring and maintaining regularity is positively reinforcing ("rewarding"), (2) the disruption of regularity is punishing, and (3) because humans tend to test their hypotheses through confirmation[28], the expected feedback serves as an incentive. Thus, we dubbed the algorithm RaCaS (Regularity as Carrot and Stick, see Fig. 1b for an illustration of RaCaS' core ideas and Fig. 1c, d for examples).

RaCaS implemented the above premises in the following manner (see "Methods" for a more formal description of the algorithm). First, rewards were transiently allocated, in a repeating sequence, only to the Bias+ option. With successive choices of the Bias+ option, rewards were allocated to that option in predictable and incrementally increasing intervals (see Fig. 1d; green-shaded trials). This regularity continued to unfold as long as the participant continued to choose the Bias+ option (premise 1, above), but if the participant explored the alternative

(Bias−) option, regularity broke down and reappeared only after the participant re-committed to the Bias+ option (premise 2; see Fig. 1c; yellow shading). Multiple rewards that were necessarily allocated to the Bias− option to comply with the CEC rules, were allocated exactly when persevering with the Bias+ option was assumed to confirm the participants' hypotheses about the existence and nature of unfolding regularity (premise 3). Any residual rewards allocated to the Bias− option were delayed until the final trials of the experiment (see Fig. 1c; orange shading). The implementation differed slightly across participants since it was dictated by the specific history of choices and rewards, as well as the current trial within a session (for the complete description of the algorithm, including all edge cases, see the publicly available online[29]).

## Results
### An effective nudge
We submitted two versions of the RaCaS algorithm to the CEC's Dynamic track. Both were based on the principles discussed above and were implemented in an almost identical manner (for the designed difference between them see Supplementary Fig. 5). In fact, the two versions of RaCaS replicated each other's results (see Fig. 2a and Supplementary Fig. 1), including the bias they induced (this and all following tests are two-tailed; mean Bias+ of 70.4% vs. 69.2%; $t(260) = 0.563$, $p = 0.573$, two samples $t$-test; 95% CI: −3%–5%). To quantify support for the null hypothesis, we conducted a Bayesian two samples $t$-test. This analysis produced a Bayes Factor (BF01) of 6.35 which is conventionally considered as moderate evidence for the lack of a difference between both versions of RaCaS.

Both versions performed better than all other algorithms submitted to the CEC, regardless of whether they used participants' previous choices to bias behavior or not (i.e., dynamic and static schedules, "DS" or "SS" respectively, see Fig. 2a). DS01 vs. SS01 $t(730) = 4.758$, $p < 0.001$, 95% CI: 3%–8%, $t$-test, and DS02 vs. SSO1 $t(730) = 3.749$, $p < 0.001$, 95% CI: 2%–7%, two samples $t$-test). We henceforth refer to the data pooled across the DS1 and DS2 algorithms as RaCaS.

Averaging 69.8% for the Bias+ option, RaCaS strongly influenced participants' choices, making them choose the Bias+ option twice as frequently as the Bias− option on average, across all participants and trials (see Fig. 2). These results reflect a large standardized effect size (Cohen's $d = 1.16$).

### Uneven gains as a self-reinforcing negative feedback loop
Computational models of decision-making characteristically estimate the value of an option as a function of its reward history. To quantify the difference in rewards received in the Bias+ vs. the Bias− option, we used a normalized-per-participant measure $\Delta_{\text{rewards}}(\text{norm.}) = \frac{\text{bias}^+_{\text{rewards}} - \text{bias}^-_{\text{rewards}}}{\text{bias}^+_{\text{rewards}} + \text{bias}^-_{\text{rewards}}}$. By dividing the reward difference between sides by the sum of rewards on both sides, this quantity ranges from 1 (all rewards received on the Bias+ side) to −1 (all rewards received on the Bias− side), with 0 for equality. The measure reflects the relative difference, as experienced by the participant (using other measures such as the raw deltas yielded similar results to those reported below, see Supplementary Data 1).

As RaCaS induced a preference for the Bias+ side, finding that the distribution of the rewards participants uncovered was unbalanced in favor of that side was trivial, since it follows that more rewards would be earned in the option that was selected more frequently. We indeed found that for RaCaS the average $\Delta_{\text{rewards}}(\text{norm.})$ measure (0.49) was significantly higher than chance (Fig. 3a, b and Supplementary Data 1, $t(261) = 14.36$, $p < 0.001$, 95% CI: 0.022–0.070, one sample $t$-test) and, echoing choice-bias, was the highest of all algorithms. This pattern was also consistent across participants, and stable throughout trials (Fig. 3c, d).

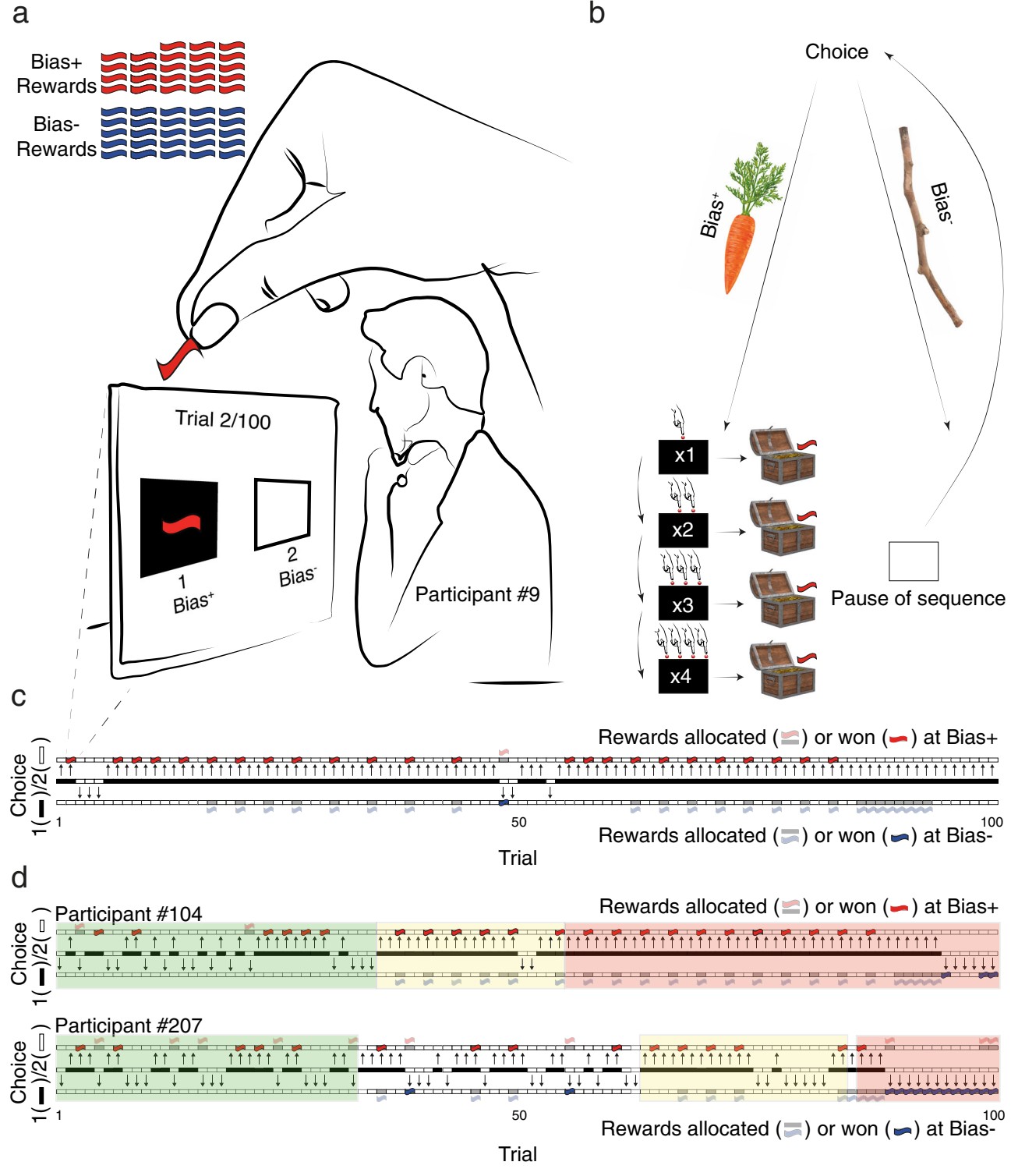

**a**

**b**

**c** Rewards allocated or won at Bias+; Rewards allocated or won at Bias-

**d** Participant #104; Participant #207; Rewards allocated or won at Bias+; Rewards allocated or won at Bias-

Less trivially, the pull of regularity seemed to also instigate a self-reinforcing negative feedback loop that further diminished the selection of the Bias⁻ option. The participants' tendency to progressively under-sample the Bias⁻ option enabled the algorithm to "hide" many of the rewards that were (obligatorily) allocated to the Bias⁻ side (Supplementary Fig. 2) such that even when participants sampled that side, they often failed to find a reward. This, in turn, further accentuated the imbalance in rewards earned between the Bias⁺ and the Bias⁻ options.

Quantitatively, this latter imbalance can be captured by accounting for the rewards earned when selecting an option divided by the number of times that option was chosen, or participants'

observed expectancy ($\frac{\text{\#Rewards won(option)}}{\text{\#choices(option)}}$). From the participants' perspective, the Bias⁻ option (which had an identical objective expectancy of winning rewards) was empirically "proven" to be less beneficial on average (see Supplementary Data 1 for the comparison to chance level and Supplementary Notes, the principle of "Learned helplessness"). On average, this held throughout the session (Fig. 3e; see also the average across trials, Supplementary Fig. 3).

Hence, this "vicious circle" promoted a false estimate of reality established by incomplete evidence, that effectively nullified the equal conditions between the options. This underestimation of the expected value of Bias⁻ likely further increased the bias away from that option.

**Fig. 1 | The Choice Engineering Competition.** The participant's screen, the algorithm's behavior, and three participants' data. **a** Participants made 100 choices between two options (here: the black and white "sides"). Unbeknownst to them, a single algorithm determined when and on which side rewards would be allocated. **b** A schematic of the RaCaS algorithm: choosing Bias+ option leads to a structured sequence of rewards, with increasing intervals (1 ≤ interval ≤ 4) between rewards. Choosing the Bias− option initially led to no reward and also punished the participants by suspending the regularity associated with Bias+. Symbols contributors: hadkhanong (Carrot), Africa Studio (Stick), meen_na (Red button), Francesco Milanese (Treasure box)−stock.adobe.com. **c** All choices made by a participant strongly biased towards the Bias+ option (participant #9; participant number reflects their ranking as a function of strength of bias). Choice is indicated by the color of the central strip (black for Bias+ and white for Bias−) and the arrow pointing up towards Bias+ or down towards Bias−. Rewards allocated by RaCaS are represented by the gray bar; a colored tilde indicates when the participant collected the reward (blank when missed). Note the developing regularity in rewards allocated to Bias+ as long as the participant showed commitment to the option (i.e., when the middle line is black for multiple consecutive trials). Intervals between rewards increase every few iterations, further luring participants to uncover (and then confirm) the new pattern. The actual allocation depended on the participant's behavior and the number of remaining trials and rewards. **d** Choices by two other participants with medium (participant #104) and weak biases (participant #207); the different events generated by the algorithm are highlighted. Green shading: the initial period, in which (structured) rewards are often allocated to Bias+, and not at all to the Bias− side, to encourage the participant to further explore Bias+. Yellow shading: the pattern begins to unfold and exploration of the Bias− option terminates the pattern. The pattern only begins anew when recommitment is detected. Orange shading: the Bias− rewards are largely hidden, since they are only allocated when the participant is highly committed to the Bias+ side until the residual rewards must be allocated throughout the remaining trials.

## The cost of sticking to regularity

Unpacking the standardized $\Delta_{rewards}$(norm.) measure we find that the unbalanced distribution of reward discovery that characterized RaCaS was caused by both the high discovery rate of rewards on the Bias+ side, and the low rate of rewards discovered on the Bias− side (see Supplementary Fig. 2). In other words, the decision-makers preference for regularity led them to stick with the Bias+ option and hence to discover fewer rewards allocated to the Bias− option.

This raises the question of whether biased participants and specifically, RaCaS assigned participants, earned fewer rewards in total (Fig. 4a–c). As a general trend, we find that "open-minded" participants −those whose choice was the least biased to either side−earned more on average (see Fig. 4a, left; the red rectangle bounds the middle band of participants with a bias of 0.4–0.6 vs. the others). The more biased they were to either side, the lower the participants' gains (Fig. 4a, right and left, Spearman's $r(260) = -0.346$, $p < 0.001$).

Notably, although RaCaS outperformed all the other competitors in terms of the bias it induced it was one of the lower-ranked algorithms in terms of the payoff won by the participants, who averaged no more than chance level winnings (see Supplementary Data 1 and Fig. 4a–c, $t(261) = 1.689$, $p = 0.09$, 95% CI: 24.93–25.82, one-sample $t$-test). To quantify support for the null hypothesis, we conducted a Bayesian one sample $t$-test. This analysis produced a Bayes factor (BF01) of 3.55, which is conventionally considered as an anecdotal to moderate evidence for the lack of a difference between earnings under RaCas and chance level earnings. RaCaS led unsuspecting individuals to forego payoffs for maintaining a specious regularity as shown in participants' consistently low ratio of discovered rewards allocated to the Bias− side ("Exploitation", defined as $\frac{\#Rewards\ won(option)}{\#Rewards\ allocated(option)}$, Fig. 4d).

Since the rewards allocated to the biased− side were often abundant during the final trials, participants who were not strongly attached to regularity, and thus more willing to explore at that later stage, were able to change their policy on these trials and discover these late rewards (for example, see Fig. 1c, d: note the last batch of trials of the three participants ranked in decreasing order according to their bias from top to bottom). This tendency to explore less due to the attraction of regularity was also reflected in the shorter Response Times (RT) of RaCaS participants when choosing the Bias+ option as compared to when choosing the Bias− option. This difference in RT was correlated with the bias (Supplementary Fig. 4, Spearman's $r(259) = -0.38$, $p < 0.001$. Note that response time was missing for one participant).

## Discussion

The bias generated by RaCaS is evidence that the mere existence of structure (in this case−structured sequences of rewards in the Bias+ side) attracts decision makers, even as they attempt to maximize their gains. Nevertheless, the question of what exactly in the regularity led participants to prefer the Bias+ option needs to be further explored.

Structure allows for predictability. Thus, the increase in predictability may have led the participants to prefer the structured option. Predictability figures in multiple psychological and neuroscientific frameworks, often as an intrinsically desired or reinforcing state. For example, the motivation for perceiving the world as orderly may drive the belief that the world is just, thus shaping the blame and praise of others[30]; and in the social psychological framework of System Justification, the predictability of a social system (e.g., political) is one of the reasons individuals are most motivated to maintain it[31]. Moreover, prediction itself has been put forward as a key driver of neural computation ("predictive coding"[32,33]).

Other psychological theories have suggested that predictions may reinforce more locally and may be related to the concept of "agency". For example, we (e.g., refs. [34,35]) previously demonstrated that (effective) movement is reinforced when it leads to accurate sensorimotor predictions (i.e., predictions about the sensations that will follow from the execution of a movement). This case can be seen as a refinement of the more general concept of control, where having a choice and influencing one's environment are considered desirable[36,37] whereas their loss is considered aversive[38].

Overall, it is difficult to determine whether regularity attracted the participants by increasing their feeling of predictability, their control[39], or alternatively, that correct predictions directly reinforced behavior. Furthermore, multiple other factors have been proposed[40] to influence decision making in similar tasks. A limitation of the current study is the inability to disambiguate between these different psychological mechanisms (e.g., predictability, control, direct reinforcement), which could lead to different theoretical interpretations. For example, risk or ambiguity aversion were possibly mitigated by the structured environment associated with the Bias+ option and could also account for RaCaS' success.

The current study did not directly manipulate or assess a number of alternative motivational factors stated above, such as risk aversion or an experience of agency. The absence of targeted manipulations means that the influence (if any) of these factors remains speculative, and future research should design experiments that will systematically isolate and assess the relative contribution of these different factors. Additionally, future research where, for example, participants are explicitly asked about the amount of control they felt during the task, or their confidence in when rewards will be received, as well as their ability to explicitly reproduce the observed pattern, while using algorithms that systematically differ in the degree of regularity of the rewards allocated, could help tease these possibilities apart.

As mentioned above, recent reviews of the efficacy of nudges suggest that, to date, little is known about how or if people's choices can be substantially affected by subtle or concealed interventions.

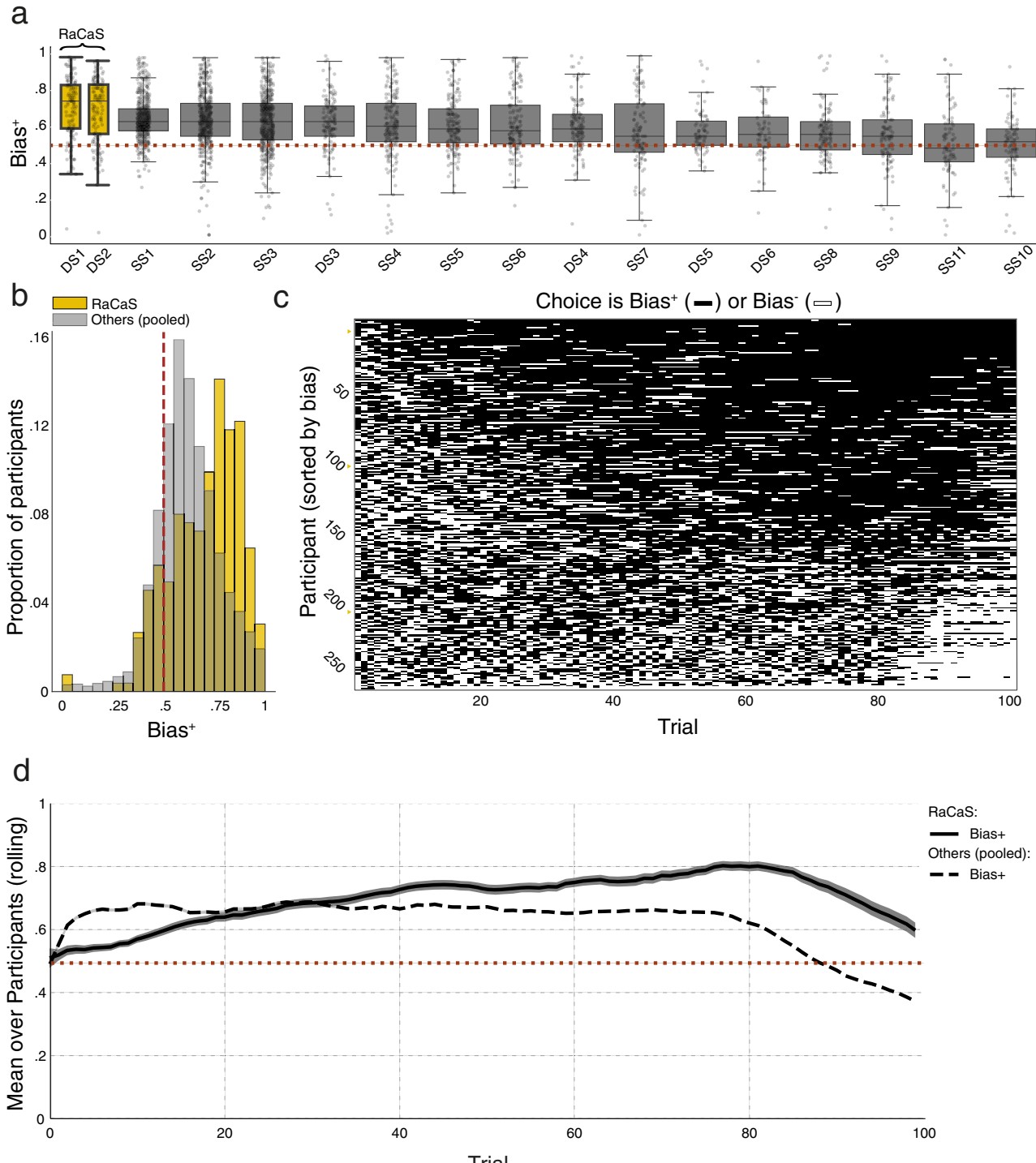

**Fig. 2 | The bias-generating performance of the RaCaS algorithm. a** Boxplots depicting the performance of all the algorithms submitted to the CEC, ordered by magnitude of bias they created. RaCaS is depicted in yellow. Dots indicate individual participants. DS dynamic schedule, SS static schedule. **b** Two overlaid histograms depicting the proportion of participants showing the magnitude of bias, with 1 denoting maximal bias. RaCaS' performance is shown in yellow and all other competing algorithms appear in gray. The brown dashed line depicts indifference between options. **c** A raster plot depicting all the RaCaS participants' choices across all 100 trials (the full dataset, ordered by the strength of bias such that the more bias exhibited by a participant, the higher the row depicting their data). A small black bar depicts a single choice of Bias+ and a small white bar, a single choice of Bias−. Small yellow triangles point to the rankings of the three participants depicted in Fig. 1c, d. **d** The dynamics of the average of Bias+ produced by RaCaS (solid line) and all other competing algorithms (dashed line) across all 100 trials. The RaCaS average bias increased throughout most of the experiment, and always remained well above chance level. The rolling average was calculated with a temporal window of size 10. Bands indicate SEM. Box bounds show interquartile range (IQR), line represent the median, and whiskers extend to points that lie within 1.5 IQRs.

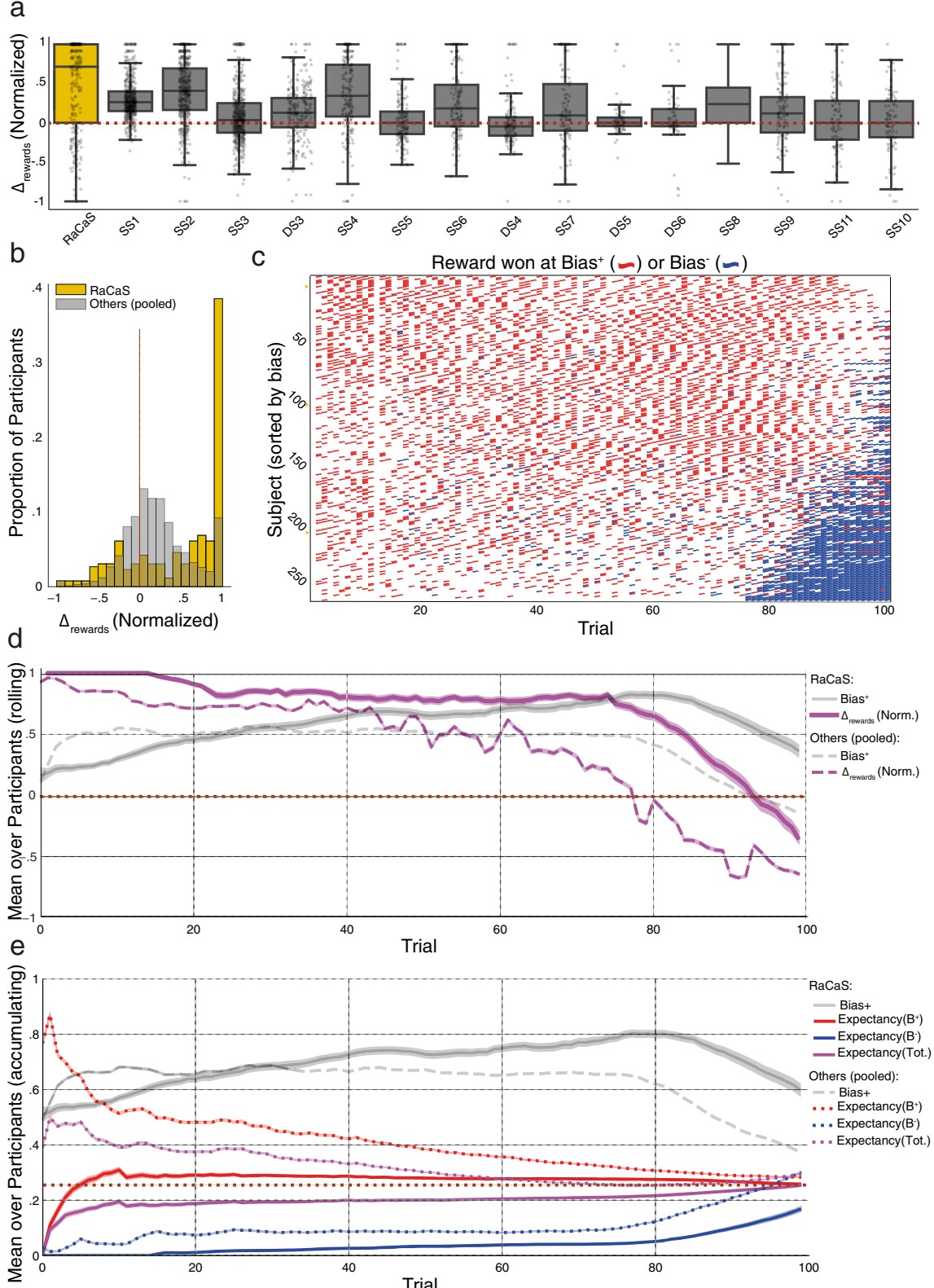

Nevertheless various actors have much to gain from devising such interventions[26,41], and may in fact take advantage of the belief that they are ineffective and hence harmless[26]. The truth is that we do not know, yet given the scant data on potentially harmful nudges, malevolent actors may appear to meet the most stringent of regulations and still intentionally and adversely influence unsuspecting digital consumers or seekers of information. At the very least, the current demonstration should make it clear to regulators that subtle changes in the decision environment can have a real and negative impact on individual and societal interests.

By capitalizing on humans' preference for regularity, RaCaS biased participants' choices even as they were attempting to maximize their profit. This is crucial, especially when considered in light of current estimates regarding the (in)efficacy of nudging, at least in real

**Fig. 3 | Reward allocation, its relative discovery, and the participants' empirical expectation in the context of RaCaS. a** Box plots showing the $\Delta_{rewards}$(norm.) rewards measure. DS dynamic schedule, SS static schedule. A score of +1 means that all the rewards discovered were from those allocated to the Bias+ option while a score of −1 means that all the rewards discovered were from those allocated to the Bias− option (see main text for full details) of all competing algorithms, ranked by strength of Bias+ (i.e., bias in choice of option). RaCaS is depicted in yellow and all other algorithms are depicted in gray. The dashed brown line depicts no bias in received rewards. **b** Two overlaid histograms showing the distribution of the standardized delta rewards measure. The RaCaS results are depicted in yellow and the average performance of all other competitors is depicted in gray. The dashed vertical brown line depicts no bias in received rewards. **c** Raster plot depicting discovery of the rewards allocated to the Bias+ (red) and bias− option (blue) for all RaCaS participants, across all 100 trials. Participants are ranked by magnitude of

bias, with the strongest at the top. DS dynamic schedule, SS static schedule. **d** The dynamics of the average standardized delta rewards ($\Delta_{rewards}$(norm.)) measure produced by RaCaS (solid purple) vs. all other competing algorithms (dashed purple) across all 100 trials. For ease of comparison, the strength the Bias+ generated by RaCaS and all other algorithms is again depicted. The rolling average was calculated with a temporal window of size 10. Bands indicate SEM. **e** The dynamics of the observed expectancy ($\frac{\#Rewards\ won}{\#choices}$) for option Bias+ (red), Bias− (blue), and total (purple) under RaCaS (full line) and the average of other algorithms (dashed line). Evidence accumulates progressively throughout all 100 trials. The dashed brown line indicates expectancy under random choice. To ease comparison, Bias+ is presented in gray for both RaCaS (full line) and as the average of all other algorithms (dashed line). Error bands reflect SEM. Box bounds show interquartile range (IQR), line represent the median, and whiskers extend to points that lie within 1.5 IQRs.

world contexts. A limitation of the current study is that the controlled laboratory conditions may not fully reflect the complexity of real-world decision environments, potentially limiting the generalizability of the findings. On the other hand, while RaCaS' efficacy may decrease when deployed in such contexts, the fact that several of RaCaS' design decisions were admittingly arbitrary, leaves room for optimization and suggests that the current estimate of the bias created by RaCaS can be taken as a lower bound of its biasing potential. For example, other regularities could attract individuals even more, and using a more efficient reward allocation policy could minimize the number of cases in which the Bias+ rewards are left to the very last trials, better harnessing their potential influence.

Overall, the findings described here are somewhat inconsistent with the growing consensus in psychology and neuroscience that a rather limited set of formalized reinforcement learning models are key for predicting and explaining human choice behavior[27,42,43]. More specifically, some results presented here, such as the seeming domination of the existence of structure over expected value, cannot be currently accommodated by such models (for other results that seem to be inconsistent with the predictions of RL models see Supplementary Materials).

The algorithm described above was an intuitive attempt to utilize a qualitative understanding of a human tendency; i.e., the attraction to regularity. Since it outperformed all other competing algorithms in the CEC, RaCaS contributes to illustrating that qualitative principles can be algorithmized to influence behavior. More generally, the current results are a reminder that despite the power of formal descriptions, we are still a far cry from a thorough understanding of what pushes, pulls, or attracts individuals. As such, one should not stop exploring various psychological influences on decision-making regardless of their amenability to formalization[44] or fit with currently dominant models[45].

From a decision-science perspective, these results are in line with recent suggestions[46] that for the field to advance substantially, more data are required. These should preferably come from experimental designs that include interventions harnessing multiple human tendencies in a principled manner while varying their combination[27,47,48]. Measuring their relative effectiveness in similar and tightly regulated environments can serve to empirically determine the impact of such tendencies on human behavior, as well as their sensitivity to context and malleability. This in turn could contribute to advances in the development and testing of increasingly sophisticated and accurate models of human decision-making and behavior.

In summary, both scientific interests and societal reasons converge to call for exploratory research using highly-controlled environments such as the well-defined CEC task, with potential adaptations to real-world scenarios (for example, malevolent or benevolent nudges designed to influence people's outcomes or information consumption). This effort will allow for the discovery, characterization, and

quantification of currently unknown or currently unquantified factors the drive human preference and behavior.

## Methods
### Data collection
The data for this study were collected by the organizers of the Choice Engineering Competition (CEC). The methods for collecting the data are published in three key sources, listed chronologically: the competition announcement[27], the competition website[49] and a 2023 preprint describing the competition's static track[50]. Here, we provide a stand-alone description of the methods that are relevant to the current study.

### Ethical compliance statement
The study was approved by the Hebrew University Committee for the Use of Human participants in Research, as per the procedures established by the CEC organizers. All participants provided informed consent prior to their involvement in the study. We did not collect any further data beyond what was specified by the CEC organizers, and thus did not seek additional ethical approval. The study complied with all ethical regulations for research involving human participants.

### Participant recruitment and compensation
Participants were recruited through the online labor platform Amazon Mechanical Turk. A total of 3259 participants took part in the study. No demographic information, such as age, sex, or gender, was collected from participants, and as such, sex and gender considerations were not included in the study design. Participants received a fixed participation fee of $0.40, in addition to a bonus of 1¢ for each reward they obtained throughout the experiment. Informed consent was obtained from all participants, ensuring their understanding of the experimental procedures and compensation. For the purpose of the current research, the data from all participants that appeared in the original dataset published by the CEC organizers were analyzed. Exclusion of participants with extreme response behaviors (i.e., with more than 95% of choices towards either side) does not significantly affect the primary outcomes we reported.

### Experimental procedure
Following their recruitment on Amazon Mechanical Turk, participants were randomly assigned to different experimental conditions (i.e., reward schedules). Participants then completed a repeated two-alternative forced-choice task, with the objective of accumulating as many rewards as possible. Each participant made 100 decisions, choosing between two available options, and received immediate feedback regarding whether they had obtained a reward on each trial. The method of Successive Rejects was employed to allocate additional participants to the better-performing reward schedules, ensuring sufficient sample sizes for each tested schedule.

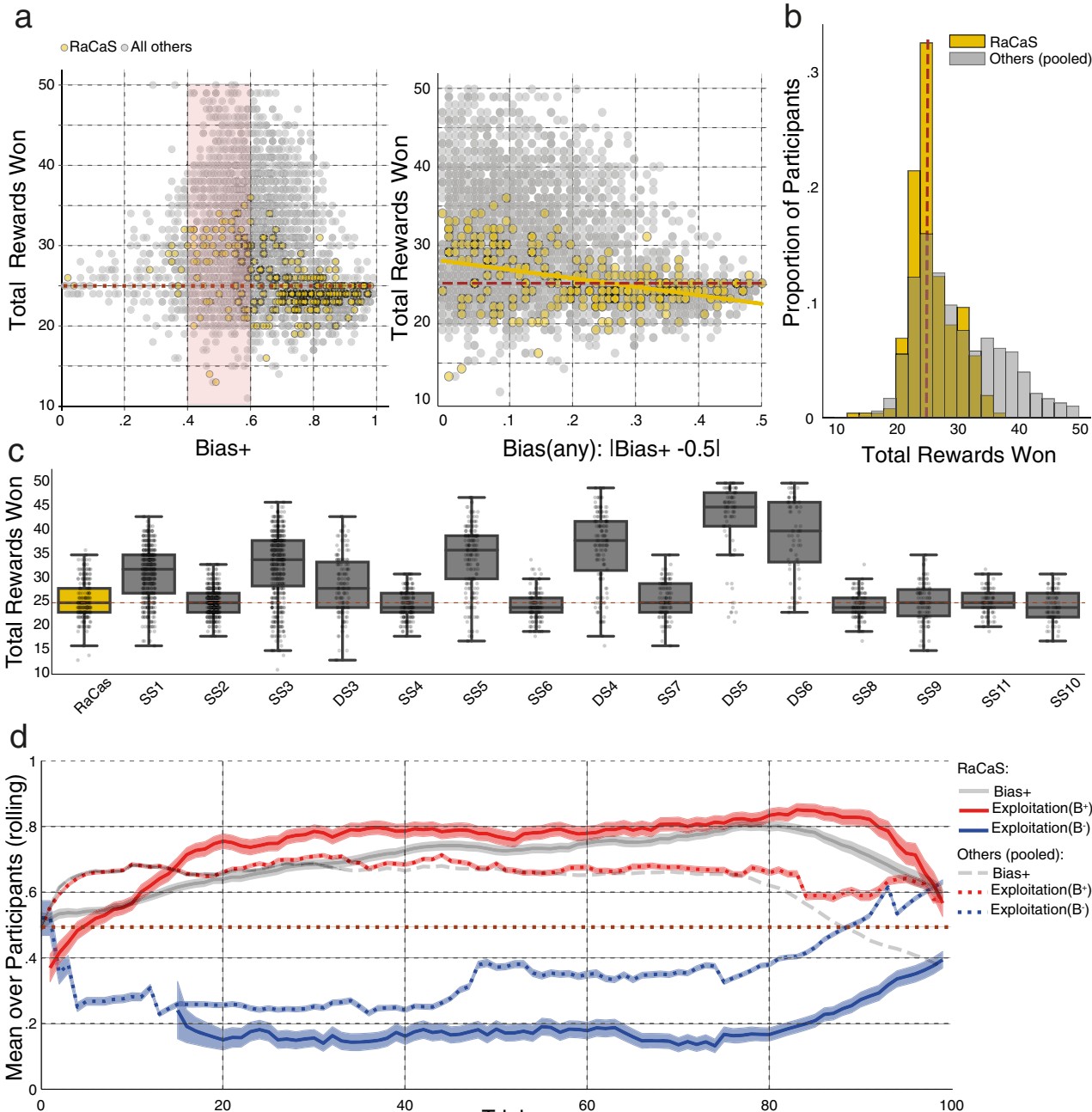

**Fig. 4 | Cost of bias for participants in the context of RaCaS and all other competing algorithms.** DS dynamic schedule, SS static schedule **a** Left: overlaid scatter plots depicting total number of obtained rewards (Y-axis) over the strength of Bias+ (X-axis). Participants under RaCaS are depicted in yellow and those under all other competing algorithms are depicted in gray. both scatters show that the participants that are the least biased towards either option earn more (comparison of the points within the red rectangle to all other data points (RaCaS; 28 vs. 24.6 rewards, $t(255) = 6.74$, $p < 0.001$, 95% CI: 2.37–4.331. Others; 30.6 vs. 28.6, $t(3257) = 7.99$, $p < 0.001$, 95% CI: 1.48–2.44. one-sample $t$-tests). Right: the X-axis reflects absolute (unsigned) bias−the amount of deviation from indifference. The regression line for RaCaS ($r(260) = -0.346$, $p < 0.001$, Spearman correlation) is overlaid on the individual observations; RaCaS participants are depicted in yellow. **b** Two overlaid histograms showing the distribution of the sum of rewards obtained by participants allocated to RaCaS (yellow) and all other algorithms (gray). The dashed vertical brown line depicts the expected sum of rewards given random choice. **c** Box plots depicting the sum of rewards obtained by participants under RaCaS (yellow) and under all other algorithms (gray). Dots represent individual participants. Algorithms are ranked by the magnitude of Bias+ they produced. The dashed brown line depicts the expected number of rewards under purely random choice. **d** The dynamics of the proportion of rewards obtained ("exploited") from those allocated to option Bias+ (red) and Bias− (blue) under RaCaS (solid line) and the pool of all other algorithms (dashed line), across all 100 trials. To ease comparison, Bias+ is again presented in gray for both RaCaS (solid line) and as the average of all other algorithms (dashed line). The rolling average was calculated with a temporal window of size 10. Error bands reflect SEM. Box bounds show interquartile range (IQR), line represent the median, and whiskers extend to points that lie within 1.5 IQRs.

## Choice Engineering Competition setup

The CEC aimed to assess how different reward schedules could influence participants' choices in a repeated decision-making task. Researchers submitted reward allocation algorithms to bias participants' preferences towards a designated target option. The competition included a Static and a Dynamic track: in the Static track, rewards were allocated based on a predetermined sequence, whereas in the Dynamic track, reward allocation adjusted in response to

participants' ongoing decisions. Rewards were equally distributed between the two options (referred to as Bias+ and Bias−), with 25 rewards assigned to each side. The goal was to covertly influence participants to prefer the Bias+ side by manipulating the timing and distribution of rewards.

### The RaCaS algorithm

RaCaS (Regularity as Carrot and Stick) was designed to harness participants' preference for predictable and regular sequences to influence their decision-making. Unlike static schedules, which did not adapt based on participants' actions, RaCaS dynamically adjusted the reward sequence based on participants' choices, thus maintaining engagement and enhancing its influence on behavior.

RaCaS implemented a phased reward allocation mechanism that evolved over six distinct stages during the 100 trials. During the initial phase (Stage 0), rewards were assigned to the Bias+ option in an alternating OFF-ON pattern contingent on the participant making a non-rewarded click first, and then being rewarded on the next click. This pattern aimed to build commitment to the Bias+ side early on by establishing predictability. In subsequent stages, the number of consecutive clicks required to receive a reward increased, thus maintaining a balance between engagement and challenge. The six stages of RaCaS were defined as follows:

- Stage 0 (Trials 1–11): every second click on Bias+ resulted in a reward ($X = 1$).
- Stage 1 (Trials 12–25): every third click on Bias+ was rewarded ($X = 2$).
- Stage 2 (Trials 26–40): every fourth click on Bias+ was rewarded ($X = 3$).
- Stage 3 (Trials 41–60): every fifth click on Bias+ was rewarded ($X = 4$).
- Stage 4 (Trials 61–80): every fourth click on Bias+ was rewarded again ($X = 3$).
- Stage 5 (Trials 81–100): every third click on Bias+ was rewarded ($X = 2$).

If participants veered away from the Bias+ option by choosing Bias−, the evolving reward sequence for Bias+ would halt, effectively breaking the pattern and introducing uncertainty. This mechanism was designed to discourage exploration of the Bias− option through (epistemic) punishment and to emphasize the (epistemic) negative reinforcement gained through consistent adherence to Bias+. To further bolster commitment to Bias+, RaCaS included a reset feature, whereby ten consecutive non-rewarded clicks led to a simplified version of the reward schedule, making rewards easier to obtain and re-engage participants.

Rewards allocated to the Bias− side followed a strict criterion of being distributed only after certain conditions were met. For example, rewards on Bias− would only be granted simultaneously with a Bias+ reward or at the end of the experiment, further reinforcing the perception that the Bias+ side was the more rewarding option.

### Data processing and analysis

Data processing for this study was conducted without reliance on third-party tools. All data preprocessing and analysis were performed using Python, with a focus on the Pandas[51], Matplotlib[52], Seaborn[53], library for data handling and manipulation. Statistical analysis included calculating metrics of choice bias and reward distribution. The Bayesian analyses were performed using JASP.

### Statistical analysis

Statistical analyses were conducted to determine whether participants deviated significantly from the null hypothesis (H0), which posited that in the absence of an effective algorithm, the proportions of choices for Bias+ and Bias− should be approximately equal, with each option chosen about 50% of the time. Similarly, under the null hypothesis, reward distribution was expected to be balanced, with each participant receiving a total of 25 rewards across the 100 trials, distributed equally between the two sides.

To evaluate the biasing effect of RaCaS, metrics such as mean choice bias towards Bias+ and total rewards received were calculated. The effectiveness of RaCaS (and all other competing algorithms) was assessed by measuring deviation from H0, with statistical tests such as $t$-tests employed to compare observed outcomes against the expected balanced outcomes. Effect sizes were calculated to quantify the magnitude of any detected biases, and normalized metrics (e.g., $\Delta_{\text{rewards}}$(norm.)) were used to compare relative differences in rewards obtained between the two options (please refer to Supplementary Data 1).

### Reporting summary

Further information on research design is available in the Nature Portfolio Reporting Summary linked to this article.

## Data availability

This manuscript is based on data made available by the organizers of the Choice Engineering Competition. The CEC dataset is publicly available[49].

## Code availability

Our Python code is publicly available[29] and includes the original code used for running the RaCaS algorithm.

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

## Acknowledgements

This study was enabled by an EMBO Long-Term Fellowship (ALTF 207-2022) to H.S.N. and an Israel Science Foundation (ISF) grant No. 1714/23 and Binational Science Foundation (BSF) grant No. 2022321 to B.E. We wish to thank Tiago Branco, Mark Anderman, Jonnathan Singh, Jasmine Reggiani, Ben Jerry Gonzales, David Beniaguev, Omri Gildai and Libi Feigin for graciously reviewing and commenting on an earlier version of this manuscript. We wish to thank the CEC's organizers for the brilliant formulation of a fascinating challenge.

## Author contributions

H.S.N. and O.A. designed the CEC winning algorithm together. The study conception was jointly shaped by H.S.N., B.E., and O.A. and the computation and software development were led by O.A., with support from H.S.N. Formal analysis was conducted by H.S.N. and O.A. Data

curation was led by O.A., with support from H.S.N. H.S.N. and B.E. led the writing of the manuscript draft. The critical review and revision of the manuscript were done by H.S.N., B.E., and O.A. H.S.N. led the visualization and data presentation, with support from O.A. and B.E., H.S.N. led the supervision, with B.E. providing additional support. Project administration was also led by H.S.N.

## Competing interests

The authors declare no competing interests.
