## [Transparent Peer Review File · Nature Communications]

Using an algorithmic approach to shape human decision-making through attraction to patterns

Corresponding Author: Dr Haran Shani-Narkiss

Version 0:

Reviewer comments:

Reviewer #1

(Remarks to the Author)

This is a clearly written and interesting paper that I think can basically be accepted as is. It's somewhat unusual in that the paper is a description of an algorithm that won a "choice engineering" competition. The paper explains what the logic of the algorithm, and visualizes the results. The key idea is that people have a preference for "regularity" (i.e., easily predictable action-reward patterns). This preference attracts them to regular options even in the absence of differential reward. The algorithm exploits the preference to strongly bias actions. The approach is well-motivated based on some of the authors' previous research.

I have two relatively minor comments.

1) The authors frame their contribution in terms of the behavioral nudging literature. While I agree that the setup is a kind of nudge, it's also quite different from the complex real-world setups that are studied in the nudge literature. It's precisely this complexity that makes nudging in the real world difficult. So I think the authors are overselling their contribution a bit.

2) Perhaps this is not the place for it, but I would have liked to see a comparative analysis of other algorithms submitted to the competition. This paper shows some very coarse analyses, pooling all the other algorithms. I'm particularly interested in knowing whether there are multiple ways to engineer choices which may be comparably good and perhaps even composable to form stronger algorithms. In its current form, the paper emphasizes only a single mechanism. If our goal is to build the strongest choice engineering algorithm, it seems reasonable to consider hybrid approaches.

Reviewer #2

(Remarks to the Author)

Thank you for sending me this paper for review. In it, the authors describe their winning entry to the CEC competition. Their approach was guided by the intuitive notion that individuals would seek and exploit regularity. This has been strikingly confirmed by the results of the CEC, with their approach winning out against other approaches.

The paper is clearly written, approachable, and I think should be stimulating and of interest to a wide readership.

The paper raises many questions, e.g. with regards to what this means for the wider consensus around RL type methods. While some of these could possibly be addressed within this paper, I think these questions are best addressed by future work.

My main comment would be to expand, if possible the comparison to the other algorithms to try and firm up that it was really the regularity which drove the results. I appreciate this is how it was designed, but it would be excellent to try and probe this assertion a little more.

One minor comment was that on first reading I struggled to understand the algorithm, and remain slightly unclear on how regularity is re-detected, and how the different stages of the algorithm relate to the number of trials remaining. It would be helpful to clarify this.

Version 1:

Reviewer comments:

Reviewer #1

(Remarks to the Author)

I am satisfied with the revision.

(Remarks on code availability)

Reviewer #2

(Remarks to the Author)

Thank you for addressing all my comments. I do not have further comments and think this paper is good to publish now.

(Remarks on code availability)

Dear Editors and Reviewers,

Thank you for the time you have invested in reading and commenting on our manuscript. We have found the comments extremely helpful and did our best to take this opportunity to expand and improve the article.

In what follows, we provide a point-by-point response to the reviewers' comments, in bold.

REVIEWERS' COMMENTS

Reviewer #1:

This is a clearly written and interesting paper that I think can basically be accepted as is.

Many thanks for these encouraging words.

It's somewhat unusual in that the paper is a description of an algorithm that won a "choice engineering" competition. The paper explains what the logic of the algorithm, and visualizes the results. The key idea is that people have a preference for "regularity" (i.e., easily predictable action-reward patterns). This preference attracts them to regular options even in the absence of differential reward. The algorithm exploits the preference to strongly bias actions. The approach is well-motivated based on some of the authors' previous research.

I have two relatively minor comments.

1) The authors frame their contribution in terms of the behavioral nudging literature. While I agree that the setup is a kind of nudge, it's also quite different from the complex real-world setups that are studied in the nudge literature. It's precisely this complexity that makes nudging in the real world difficult. So I think the authors are overselling their contribution a bit.

In retrospect, we agree with the reviewer that the focus on the results' relevance to nudging practices should not have taken center stage. In the revision we have toned down this focus, most notably in changing the MS's title. We also reduced the place for this aspect in the discussion, and in multiple instances we were more attuned to the terminology, replacing “nudge” with terms like “influence”. Finally, while still discussing the results in the context of behavioral nudging, we explicitly acknowledge the reviewer’s comment, with the following sentence, now added to the main text (newly added text is in red):

“By capitalizing on humans’ preference for regularity, RaCas strongly biased participants’ choices under substantial constraints. This is crucial, especially when considered in light of current estimates regarding the (in)efficacy of nudging, at least in real world context. While RaCaS’s efficacy may similarly decrease when deployed in such contexts...”

2) Perhaps this is not the place for it, but I would have liked to see a comparative analysis of other algorithms submitted to the competition. This paper shows some very coarse analyses, pooling all the other algorithms. I'm particularly interested in knowing whether there are multiple ways to engineer choices which may be comparably good and perhaps even composable to form stronger algorithms. In its current form, the paper emphasizes only a single mechanism. If our goal is to build the strongest choice engineering algorithm, it seems reasonable to consider hybrid approaches.

We thank the reviewer for this comment. In the current submission we have provided a supplemental section which first details, and then empirically evaluates the efficacy of three principles that guided the design of the two schedules we submitted to the 'static' track of the CEC (please see new Supplementary text S1-S5 and new Supplementary figures 5 and 6). We show that these principles are influential (and potentially composable) elements for engineering choices (in some environments more than others, as we also discuss).

We further utilize the empirical evaluation of the above design principles to compare whether their contribution to generating a bias in participants' choice behavior changes under RaCaS, leading to some interesting insights. We believe that this strategy both strengthened the basic science aspect of the manuscript as well as provides new directions for the design of effective choice engineering environments. Notably, we also explicitly acknowledge the limitations of the current work and offer future experiments to further establish our conclusions.

Reviewer #2:

Thank you for sending me this paper for review. In it, the authors describe their winning entry to the CEC competition. Their approach was guided by the intuitive notion that individuals would seek and exploit regularity. This has been strikingly confirmed by the results of the CEC, with their approach winning out against other approaches.

The paper is clearly written, approachable, and I think should be stimulating and of interest to a wide readership.

We thank the reviewer for their kind words.

The paper raises many questions, e.g. with regards to what this means for the wider consensus around RL type methods. While some of these could possibly be addressed within this paper, I think these questions are best addressed by future work.

My main comment would be to expand, if possible the comparison to the other algorithms to try and firm up that it was really the regularity which drove the results. I appreciate this is how it was designed, but it would be excellent to try and probe this assertion a little more.

We do appreciate the reviewer's comment, which led to what we view as additional stimulating results reported in a significantly expanded Supplemental (Please see Supplemental text S1-S5 and new Supplementary figures 5 and 6). In them, we further probe the assertion that the attraction to regularity was indeed central for RaCaS's effectiveness, mostly by showing that other factors, which were key for the success of other comparable schedules, lost most or all of their effectiveness under RaCaS. Please also see our response to reviewer #1's second comment.

One minor comment was that on first reading I struggled to understand the algorithm, and remain slightly unclear on how regularity is re-detected, and how the different stages of the algorithm relate to the number of trials remaining. It would be helpful to clarify this.

We have now added a more formal and elaborated verbal description of the RaCaS algorithm. Please see the Supplemental materials, section S5.

Once again, we would like to thank the reviewers for their insights, which both encouraged us and greatly helped us in further developing and improving the manuscript.